# Decentralized Feature-Distributed Optimization for Generalized Linear Models

## Abstract

We consider the "all-for-one" decentralized learning problem for generalized linear models. The features of each sample are partitioned among several collaborating agents in a connected network, but only one agent observes the response variables. To solve the regularized empirical risk minimization in this distributed setting, we apply the Chambolle–Pock primal–dual algorithm to an equivalent saddle-point formulation of the problem. The primal and dual iterations are either in closed-form or reduce to coordinate-wise minimization of scalar convex functions. We establish convergence rates for the empirical risk minimization under two different assumptions on the loss funtion (Lipschitz and square root Lipschitz), and show how they depend on the characteristics of the design matrix and the Laplacian of the network.

## 1 Introduction

Let $\ell\colon \mathbb{R} \times \mathbb{R} \to \mathbb{R}_{\geq 0}$ denote a given *sample loss function* that is convex and, for simplicity, differentiable in its first argument. Given data points $(\boldsymbol{x}_1, y_1), \ldots, (\boldsymbol{x}_n, y_n) \in \mathbb{R}^d \times \mathbb{R}$ and a convex regularization function $r(\cdot)$, we consider the minimization of regularized empirical risk in generalized linear models, i.e.,

$$\min_{\boldsymbol{\theta} \in \mathbb{R}^d} \frac{1}{n} \sum_{i=1}^{n} \ell(\boldsymbol{x}_i^\intercal \boldsymbol{\theta}, y_i) + r(\boldsymbol{\theta}) \, ,$$

in a *non-standard* distributed setting where the data features, rather than samples, are distributed among $m$ agents that communicate through a connected network.

The problem can be formally stated as follows. With $\mathcal{A}_1, \ldots, \mathcal{A}_m$ denoting a partition of $[d] \stackrel{\text{def}}{=} \{1, \ldots, d\}$ into $m$ disjoint blocks, each agent $j \in [m]$ observes the *local features* $\boldsymbol{x}_{j,i} \stackrel{\text{def}}{=} (\boldsymbol{x}_i)_{\mathcal{A}_j} \in \mathbb{R}^{d_j}$ for every $i \in [n]$, where $(\boldsymbol{u})_{\mathcal{A}}$ denotes the restriction of $\boldsymbol{u}$ to the coordinates enumerated by the index set $\mathcal{A}$. Without loss of generality we may assume that each $\mathcal{A}_j$ is a set of $d_j$ consecutive indices and simply write[1]

$$\boldsymbol{x}_i = \begin{bmatrix} \boldsymbol{x}_{1,i}; & \cdots; & \boldsymbol{x}_{m,i} \end{bmatrix} \, .$$

We also denote the $n \times d_j$ *local design matrix* for agent $j \in [m]$ by

$$\boldsymbol{X}_j = \begin{bmatrix} \boldsymbol{x}_{j,1} & \cdots & \boldsymbol{x}_{j,n} \end{bmatrix}^\intercal \, ,$$

and the full $n \times d$ design matrix by

$$\boldsymbol{X} = \begin{bmatrix} \boldsymbol{X}_1 & \cdots & \boldsymbol{X}_m \end{bmatrix} = \begin{bmatrix} \boldsymbol{x}_1 & \cdots & \boldsymbol{x}_n \end{bmatrix}^\intercal \, .$$

We assume that only one of the agents, say the first agent, observes the response $(y_i)_{i=1}^n$ and the other agents only have access to their local features. There is an underlying communication network which can be abstracted by a *connected* undirected graph $G$ over the vertex set $\mathcal{V} = [m]$. If distinct agents $j$ and

---

[1]We denote the vertical concatenations using semicolons as the delimiters.

$j'$ can communicate directly, then they are adjacent in $G$ and we write $j \sim_G j'$. The *Laplacian* of the communication graph, which is central in the distributed computations of the optimization algorithms, is denoted by $\boldsymbol{L}$.

Using the shorthand

$$\ell_i(\cdot) \stackrel{\text{def}}{=} \ell(\cdot, y_i)$$

that we use henceforth to simplify the notation, we seek an approximation to the (regularized) *empirical risk minimizer*

$$\widehat{\boldsymbol{\theta}} = \underset{\boldsymbol{\theta} \in \mathbb{R}^d}{\operatorname{argmin}} \frac{1}{n} \sum_{i=1}^{n} \ell_i(\boldsymbol{x}_i^\intercal \boldsymbol{\theta}) + r(\boldsymbol{\theta}) \,. \tag{1}$$

where the regularizer $r(\cdot)$ is typically used to induce a certain structure (e.g., sparsity) in $\widehat{\boldsymbol{\theta}}$.

To solve this optimization in our distributed setting, we use a primal–dual formulation that accommodates local calculations. Specifically, with $\ell_i^*\colon \mathbb{R} \to \mathbb{R}$ denoting the *convex conjugate* of the function $\ell_i(\cdot)$, the minimization in (1) can be formulated as the saddle-point problem

$$\min_{\boldsymbol{\theta} \in \mathbb{R}^d} \max_{\boldsymbol{\lambda}_1 \in \mathbb{R}^n} \frac{1}{n} \boldsymbol{\lambda}_1^\intercal \boldsymbol{X} \boldsymbol{\theta} - \frac{1}{n} \sum_{i=1}^{n} \ell_i^*(\lambda_{1,i}) + r(\boldsymbol{\theta}) \,,$$

where $\boldsymbol{\lambda}_1 = \begin{bmatrix} \lambda_{1,1}; & \cdots; & \lambda_{1,n} \end{bmatrix}$ is the dual variable. The regularizer $r(\boldsymbol{\theta})$ might also be represented using its conjugate, making the objective of the resulting saddle-point problem linear in the primal variable $\boldsymbol{\theta}$. However, to avoid the need for the "dualization" of the regularizer, we focus on the special but important case that the regularizer is *separable* with respect to the agents. Partitioning the coordinates of the primal variable $\boldsymbol{\theta}$ according to the partitioning of the features among the agents as

$$\boldsymbol{\theta} = \begin{bmatrix} \boldsymbol{\theta}_1; & \cdots; & \boldsymbol{\theta}_m \end{bmatrix} \,,$$

with $\boldsymbol{\theta}_j \in \mathbb{R}^{d_j}$, we assume that the regularizer takes the form

$$r(\boldsymbol{\theta}) = \sum_{j=1}^{m} r_j(\boldsymbol{\theta}_j) \,, \tag{2}$$

where for each $j \in [m]$ the convex functions $r_j(\cdot)$ have a simple *proximal mapping* that is available to the $j$th agent. Giving each agent its own version of the dual variable denoted by $\boldsymbol{\lambda}_j \in \mathbb{R}^n$, we can express (1) in a form which is amenable to distributed computations as

$$\min_{\boldsymbol{\theta} \in \mathbb{R}^d} \max_{\boldsymbol{\lambda}_1, \dots, \boldsymbol{\lambda}_m \in \mathbb{R}^n} \sum_{j=1}^{m} r_j(\boldsymbol{\theta}_j) + \frac{1}{n} \boldsymbol{\lambda}_j^\intercal \boldsymbol{X}_j \boldsymbol{\theta}_j - \frac{1}{n} \sum_{i=1}^{n} \ell_i^*(\lambda_{1,i})$$
$$\text{subject to } \boldsymbol{L} \begin{bmatrix} \boldsymbol{\lambda}_1 & \cdots & \boldsymbol{\lambda}_m \end{bmatrix}^\intercal = \boldsymbol{0} \,. \tag{3}$$

The constraint involving the Laplacian simply enforces $\boldsymbol{\lambda}_j = \boldsymbol{\lambda}_j'$ for all $j \sim_G j'$. With $\langle \cdot, \cdot \rangle$ denoting the usual (Frobenius) inner product henceforth, we can use the Lagrangian form of the inner optimization to express (3) equivalently as

$$\min_{\boldsymbol{\theta} \in \mathbb{R}^d} \max_{\boldsymbol{\lambda}_1, \dots, \boldsymbol{\lambda}_m \in \mathbb{R}^n} \min_{\boldsymbol{V} \in \mathbb{R}^{n \times m}} \frac{1}{n} \langle \boldsymbol{V}^\intercal, \boldsymbol{L} \begin{bmatrix} \boldsymbol{\lambda}_1 \cdots \boldsymbol{\lambda}_m \end{bmatrix}^\intercal \rangle + \sum_{j=1}^{m} r_j(\boldsymbol{\theta}_j) + \frac{1}{n} \boldsymbol{\lambda}_j^\intercal \boldsymbol{X}_j \boldsymbol{\theta}_j - \frac{1}{n} \sum_{i=1}^{n} \ell_i^*(\lambda_{1,i})$$

$$= \min_{\boldsymbol{\theta} \in \mathbb{R}^d} \min_{\boldsymbol{V} \in \mathbb{R}^{n \times m}} \max_{\boldsymbol{\lambda}_1, \dots, \boldsymbol{\lambda}_m \in \mathbb{R}^n} \sum_{j=1}^{m} r_j(\boldsymbol{\theta}_j) + \frac{1}{n} \boldsymbol{\lambda}_j^\intercal \boldsymbol{X}_j \boldsymbol{\theta}_j - \frac{1}{n} \sum_{i=1}^{n} \ell_i^*(\lambda_{1,i}) + \frac{1}{n} \langle \boldsymbol{V} \boldsymbol{L}, \begin{bmatrix} \boldsymbol{\lambda}_1 \cdots \boldsymbol{\lambda}_m \end{bmatrix} \rangle$$

$$= \min_{\boldsymbol{\theta} \in \mathbb{R}^d, \ \boldsymbol{V} \in \mathbb{R}^{n \times m}} \max_{\boldsymbol{\lambda}_1, \dots, \boldsymbol{\lambda}_m \in \mathbb{R}^n} -\frac{1}{n} \sum_{i=1}^{n} \ell_i^*(\lambda_{1,i}) + \sum_{j=1}^{m} r_j(\boldsymbol{\theta}_j) + \frac{1}{n} \boldsymbol{\lambda}_j^\intercal (\boldsymbol{X}_j \boldsymbol{\theta}_j + \boldsymbol{V} \boldsymbol{L} \boldsymbol{e}_j) \,, \tag{4}$$

where the second line follows from strong duality.

In Section 2 we describe the iterations based on the Chambolle–Pock primal–dual algorithm (Chambolle & Pock, 2016) to solve the saddle-point problem (4). Our main result and the assumptions under which it holds are provided in Section 3. Some numerical experiments are also provided in Section 4. Proofs of the main result can be found in Appendix A.

## 1.1 Related work

Minimization of a sum of (convex) functions is the most studied problem in distributed optimization due to its prevalence in machine learning. The most commonly considered setting in the literature is by far the *sample-distributed* setting, where each agent merely has access to one of the summands of the objective function that can be computed using the locally available samples. The literature primarily considers two different communication models. *Centralized* first-order methods have a main computing agent that aggregates the local (sub)gradient evaluations of the other agents, updates the iterate and sends it back to the other agents. Therefore, the communication time for these methods grows linearly with the *diameter* of the underlying network. In contrast, *decentralized* first-order methods do not rely on a single aggregating agent; every agent maintains and updates a local copy of the candidate minimizer through local computations and communications with its immediate neighbors, and consistency of the solution across agents is achieved either through local averaging or consensus constraints. Due to the diffusion-style nature of the iterations, the convergence rate of these methods depends on a certain notion of *spectral gap* of the communication graph. Many algorithms have been introduced for sample-distributed decentralized convex optimization; surveys of the literature can be found in (Yang et al., 2019; Gorbunov et al., 2020), and prominent references include (Johansson et al., 2008; Nedić & Ozdaglar, 2009; Wang & Elia, 2011; Zhu & Martinez, 2012; Duchi et al., 2012; Scaman et al., 2017). In general, the computation and communication complexity of these algorithms to find an $\epsilon$-accurate solution range from the "slow rate" of $O(\varepsilon^{-2}) + O(\varepsilon^{-1})$ for Lipschitz-continuous convex functions, to the "linear rate" of $O(\log(1/\varepsilon))$ for smooth and strongly convex functions. Lower bounds and (nearly) optimal algorithms for a few common objective classes are established in (Scaman et al., 2019).

The "feature-distributed" setting that we consider is studied to a lesser extent, but has found important applications such as *sensor fusion* (Sasiadek, 2002) and *cross-silo federated learning* (Kairouz et al., 2021). This setting is also relevant in *parallelized computing* to amplify the performance of resource limited computing agents in large-scale problems.

Centralized federated learning protocols, in which the agents communicate with a server, with distributed features are proposed in (Hu et al., 2019) and (Chen et al., 2020). Hu et al. (2019) proposed the FDML method and, under convexity and smoothness of the objective, established a regret bound for SGD that decays with the number of iterations $T$ at the rate of $O(1/\sqrt{T})$. It is also assumed in this result that the iterates never exit a neighborhood of the true parameter, basically imposing the strong convexity on the objective in an implicit form. Chen et al. (2020) proposed a method called VAFL, in which a server maintains a global parameter and each client operates on local features and parameters that determine the client's corresponding predictor. The clients and the server communicate in an asynchronous fashion and exchange the value of clients' predictors and the gradients of the sample loss with respect to these predictors. Under certain models of the communication delays that impose the asynchrony, a variant of stochastic gradient descent is shown to converge at a rate $O(1/T)$ under strong convexity. The performance of VAFL in the case of smooth nonconvex objectives and nonlinear predictors that are separable across the agents is also considered in (Chen et al., 2020). However, in this general setting where the guarantees are inevitably weaker, only the temporal average of the squared norm of the gradients (in expectation with respect to the SGD samples) are shown to converge at a rate $O(1/\sqrt{T})$.

The CoLa algorithm of He et al. (2018) considers a ubiquitous class of convex minimization problems in machine learning and statistics that involve *linear predictors*, in the decentralized distributed setting. Following the formulation of (Smith et al., 2018), a pair of convex programs that are dual to each other are considered in (He et al., 2018) depending on whether the data is split across the samples, or across the features. This latter setting is the closest related work in the literature to the present paper. The main step in each iteration of the CoLa algorithm is a regularized convex quadratic minimization. This minimiza-

tion step is generally nontrivial and needs to be performed by a dedicated subroutine, though the analysis accommodates subroutines that compute inexact solutions. In contrast, our convex-concave saddle point formulation of the problem leads to iterations in which every agent evaluates either a closed-from expression or a simple proximal operator, except for one agent whose computations are as simple as performing one-dimensional strongly convex minimization for each dual coordinate. Furthermore, while our algorithm achieves an accuracy of $O(1/T)$ after $T$ iterations similar to the CoLa (in the general convex setting), our convergence analysis applies to the broader class of square root Lipschitz loss functions, defined below in Section 3, that includes the usual smooth loss functions as special case (Srebro et al., 2010, Lemma 2.1).

Arablouei et al. (2015); Gratton et al. (2018) present algorithms based on ADMM for solving decentralized least-squares problems with distributed features, and establish asymptotic convergence. A feature-decentralized algorithm for logistic regression is presented in (Slavković et al., 2007), though no convergence guarantees are given.

Finally, the primal-dual algorithm we present in the next section is related to an application of the distributed saddle point algorithm of Mateos-Núñez & Cortés (2017) where the goal is minimizing a sum of functions of independent variables subject to linear inequality constraints (see Remark III.1 in that paper). The general algorithm considered in (Mateos-Núñez & Cortés, 2017) is a (projected) gradient descent ascent method. Consequently, its convergence analysis relies on certain boundedness assumptions on the iterates and the corresponding gradients. Furthermore, while being applicable to a broader set of saddle point problems than our method, this gradient descent ascent method is only shown to converge at the rate $1/\sqrt{T}$.

## 1.2 Contributions

Using the dual representation of the loss functions $\ell_i(\cdot)$ in (1), we convert the corresponding minimization problem to a saddle-point problem that enables us to perform the decentralized minimization in the unconventional feature-distributed setting.

Using the Chambole–Pock primal–dual algorithm as a natural method to solve the resulting saddle-point problem, we provide convergence guarantees for the algorithm in terms of the primal objective (rather than the primal–dual gap). In particular, we show convergence to the minimum primal value at a rate of $1/T$, assuming that the loss functions $\ell_i(\cdot)$ are either Lipschitz or square root Lipschitz smooth. The square root Lipschitz smooth functions include the more common Lipschitz gradient functions.

In each iteration, the updates to the primal variables are either in closed-form or involve evaluation of a simple proximal mapping. Updating the dual variable corresponding to the "main agent" among the $n$ agents, only requires computing the Moreau envelope of $n$ univariate functions that is highly parallelizable and can be solved efficiently. Updating the dual variables for the rest of the agents is even simpler and is expressed in closed-form.

## 2 The primal–dual algorithm

Let $f$ and $g$ be convex functions such that $f$ is smooth and has a tractable first-order oracle, and the possibly nonsmooth $g$ admits a tractable proximal mapping. Furthermore, let $h$ be a convex function whose *convex conjugate*, denoted by $h^*$, admits a tractable proximal mapping. The Chambolle–Pock primal–dual algorithm (Chambolle & Pock, 2016) solves the saddle-point problem

$$\min_{\boldsymbol{z}} \max_{\boldsymbol{\lambda}} f(\boldsymbol{z}) + g(\boldsymbol{z}) + \boldsymbol{\lambda}^\mathsf{T} \boldsymbol{K} \boldsymbol{z} - h^*(\boldsymbol{\lambda}),$$

for a given matrix $\boldsymbol{K}$. Denoting the columns of $\boldsymbol{V}$ by $\boldsymbol{v}_1, \ldots, \boldsymbol{v}_m$, and the Kronecker product by $\otimes$, the optimization problem (4) fits into the above formulation by choosing

$$\boldsymbol{z} = \begin{bmatrix} \boldsymbol{\theta}_1; & \cdots; & \boldsymbol{\theta}_m; & \boldsymbol{v}_1; & \cdots; & \boldsymbol{v}_m \end{bmatrix},$$
$$\boldsymbol{\lambda} = \begin{bmatrix} \boldsymbol{\lambda}_1; & \cdots; & \boldsymbol{\lambda}_m \end{bmatrix},$$

$$\boldsymbol{K} = \frac{1}{n} \begin{bmatrix} \boldsymbol{X}_1 & \boldsymbol{0} & \boldsymbol{0} & \cdots & \boldsymbol{0} \\ \boldsymbol{0} & \boldsymbol{X}_2 & \boldsymbol{0} & \cdots & \boldsymbol{0} \\ \vdots & \vdots & & \ddots & \vdots \\ \boldsymbol{0} & \boldsymbol{0} & \boldsymbol{0} & \cdots & \boldsymbol{X}_m \end{bmatrix} \quad \boldsymbol{L} \otimes \boldsymbol{I} \, ,$$

$$f \equiv 0 \, ,$$

$$g(\boldsymbol{z}) = r(\boldsymbol{\theta}) = \sum_{j=1}^{m} r_j(\boldsymbol{\theta}_j) \, ,$$

and

$$h^*(\boldsymbol{\lambda}) = \frac{1}{n} \sum_{i=1}^{n} \ell_i^*(\lambda_{1,i}) \, .$$

The update rule of the Chambolle–Pock algorithm can be summarized as

$$\boldsymbol{z}_{t+1} = \operatorname*{argmin}_{\boldsymbol{z} \in \mathbb{R}^{d+mn}} \; f(\boldsymbol{z}_t) + \langle \nabla f(\boldsymbol{z}_t), \boldsymbol{z} - \boldsymbol{z}_t \rangle + g(\boldsymbol{z}) + \boldsymbol{\lambda}_t^\mathsf{T} \boldsymbol{K} \boldsymbol{z} + \frac{1}{2\tau} \|\boldsymbol{z} - \boldsymbol{z}_t\|_2^2$$

$$\boldsymbol{\lambda}_{t+1} = \operatorname*{argmin}_{\boldsymbol{\lambda} \in \mathbb{R}^{mn}} \; h^*(\boldsymbol{\lambda}) - \boldsymbol{\lambda}^\mathsf{T} \boldsymbol{K} \left(2\boldsymbol{z}_{t+1} - \boldsymbol{z}_t\right) + \frac{1}{2\sigma} \|\boldsymbol{\lambda} - \boldsymbol{\lambda}_t\|_2^2 \, ,$$

for appropriately chosen parameters $\tau, \sigma > 0$. Writing this update explicitly for our special case, we have

$$\boldsymbol{z}_{t+1} = \operatorname*{argmin}_{\boldsymbol{z} \in \mathbb{R}^{d+mn}} \; r\left((\boldsymbol{z})_{[d]}\right) + \boldsymbol{\lambda}_t^\mathsf{T} \boldsymbol{K} \boldsymbol{z} + \frac{1}{2\tau} \|\boldsymbol{z} - \boldsymbol{z}_t\|_2^2$$

$$\boldsymbol{\lambda}_{t+1} = \operatorname*{argmin}_{\boldsymbol{\lambda} \in \mathbb{R}^{mn}} \; \frac{1}{n} \sum_{i=1}^{n} \ell_i^*(\lambda_{1,i}) - \boldsymbol{\lambda}^\mathsf{T} \boldsymbol{K} (2\boldsymbol{z}_{t+1} - \boldsymbol{z}_t) + \frac{1}{2\sigma} \|\boldsymbol{\lambda} - \boldsymbol{\lambda}_t\|_2^2 \, .$$

Expanding the linear term in the primal update, the equivalent local primal update for each agent $j \in [m]$ can be written as

$$\boldsymbol{\theta}_{j,t+1} = \operatorname*{argmin}_{\boldsymbol{\theta}_j \in \mathbb{R}^{d_j}} \; r_j(\boldsymbol{\theta}_j) + \frac{1}{n} \boldsymbol{\lambda}_{j,t}^\mathsf{T} \boldsymbol{X}_j \boldsymbol{\theta}_j + \frac{1}{2\tau} \|\boldsymbol{\theta}_j - \boldsymbol{\theta}_{j,t}\|_2^2, \tag{5}$$

$$\boldsymbol{v}_{j,t+1} = \operatorname*{argmin}_{\boldsymbol{v}_j \in \mathbb{R}^n} \; \frac{1}{n} \left( \sum_{j' \in [m] : \, j \sim_G j'} \boldsymbol{\lambda}_{j,t} - \boldsymbol{\lambda}_{j',t} \right)^\mathsf{T} \boldsymbol{v}_j + \frac{1}{2\tau} \|\boldsymbol{v}_j - \boldsymbol{v}_{j,t}\|_2^2. \tag{6}$$

Similarly, the equivalent local dual update for each agent $j \in [m] \backslash \{1\}$ is

$$\boldsymbol{\lambda}_{j,t+1} = \operatorname*{argmin}_{\boldsymbol{\lambda}_j \in \mathbb{R}^n} \; -\frac{1}{n} \boldsymbol{\lambda}_j^\mathsf{T} \left( \sum_{j' \in [m] : \, j \sim_G j'} 2(\boldsymbol{v}_{j,t+1} - \boldsymbol{v}_{j',t+1}) - \boldsymbol{v}_{j,t} + \boldsymbol{v}_{j',t} \right)$$
$$- \frac{1}{n} \boldsymbol{\lambda}_j^\mathsf{T} \boldsymbol{X}_j \left(2\boldsymbol{\theta}_{j,t+1} - \boldsymbol{\theta}_{j,t}\right) + \frac{1}{2\sigma} \|\boldsymbol{\lambda}_j - \boldsymbol{\lambda}_{j,t}\|_2^2 \, . \tag{7}$$

The fact that $h^*(\cdot)$ depends entirely on $\boldsymbol{\lambda}_1$ makes the local dual update for the first agent (i.e., $j = 1$) different and in the form

$$\boldsymbol{\lambda}_{1,t+1} = \operatorname*{argmin}_{\boldsymbol{\lambda}_1 \in \mathbb{R}^n} \; \frac{1}{n} \sum_{i=1}^{n} \ell_i^*(\lambda_{1,i}) - \frac{1}{n} \boldsymbol{\lambda}_1^\mathsf{T} \left( \sum_{j' \in [m] : \, 1 \sim_G j'} 2(\boldsymbol{v}_{1,t+1} - \boldsymbol{v}_{j',t+1}) - \boldsymbol{v}_{1,t} + \boldsymbol{v}_{j',t} \right)$$
$$- \frac{1}{n} \boldsymbol{\lambda}_1^\mathsf{T} \boldsymbol{X}_1 \left(2\boldsymbol{\theta}_{1,t+1} - \boldsymbol{\theta}_{1,t}\right) + \frac{1}{2\sigma} \|\boldsymbol{\lambda}_1 - \boldsymbol{\lambda}_{1,t}\|_2^2 \, , \tag{8}$$

where the scalars $(\lambda_{1,i})_i$ denote the coordinates of $\boldsymbol{\lambda}_1$ and should not be confused with the vectors $(\boldsymbol{\lambda}_{1,t})_t$. The primal update (5) is simply an evaluation of the proximal mapping of $\tau r_j$ denoted by $\operatorname{prox}_{\tau r_j}(\boldsymbol{u}) =$

$\arg\min_{\boldsymbol{u}'} \tau r_j(\boldsymbol{u}') + \|\boldsymbol{u}' - \boldsymbol{u}\|_2^2/2$. The updates (6) and (7) can also be solved in closed-form. While (8) does not admit a similar closed-form expression, it can be equivalently written in terms of the functions $\ell_1(\cdot), \ldots, \ell_n(\cdot)$ using the separability of the objective function and the relation between the *Moreau envelope* of a function and its convex conjugate (Bauschke & Combettes, 2011, Proposition 13.24). Therefore, we can summarize the iterations as

$$\boldsymbol{\theta}_{j,t+1} = \text{prox}_{\tau r_j}\left(\boldsymbol{\theta}_{j,t} - \frac{\tau}{n}\boldsymbol{X}_j^\intercal \boldsymbol{\lambda}_{j,t}\right), \quad \text{for } j \in [m], \tag{9}$$

$$\boldsymbol{v}_{j,t+1} = \boldsymbol{v}_{j,t} - \frac{\tau}{n}\sum_{j'\in[m]:\, j\sim_G j'} \boldsymbol{\lambda}_{j,t} - \boldsymbol{\lambda}_{j',t}, \quad \text{for } j \in [m], \tag{10}$$

$$\boldsymbol{\lambda}_{j,t+1} = \boldsymbol{\lambda}_{j,t} + \frac{\sigma}{n}\boldsymbol{X}_j\left(2\boldsymbol{\theta}_{j,t+1} - \boldsymbol{\theta}_{j,t}\right) + \frac{\sigma}{n}\sum_{j'\in[m]:\, j\sim_G j'} 2(\boldsymbol{v}_{j,t+1} - \boldsymbol{v}_{j',t+1}) - \boldsymbol{v}_{j,t} + \boldsymbol{v}_{j',t}, \quad \text{for } j \in [m]\backslash\{1\}, \tag{11}$$

$$\boldsymbol{\lambda}_{1,t+1} = \underset{\boldsymbol{\lambda}_1 \in \mathbb{R}^n}{\arg\min} \frac{1}{n}\sum_{i=1}^n \ell_i\left(\frac{n}{\sigma}\left(\boldsymbol{\lambda}_{1,t+1/2} - \boldsymbol{\lambda}_1\right)_i\right) + \frac{1}{2\sigma}\|\boldsymbol{\lambda}_1\|_2^2, \tag{12}$$

where $(\boldsymbol{u})_i$ denotes the $i$th coordinate of a vector $\boldsymbol{u}$, and the "intermediate dual iterate" $\boldsymbol{\lambda}_{1,t+1/2}$ is defined as

$$\boldsymbol{\lambda}_{1,t+1/2} = \boldsymbol{\lambda}_{1,t} + \frac{\sigma}{n}\boldsymbol{X}_1\left(2\boldsymbol{\theta}_{1,t+1} - \boldsymbol{\theta}_{1,t}\right) + \frac{\sigma}{n}\sum_{j'\in[m]:\, 1\sim_G j'} 2(\boldsymbol{v}_{1,t+1} - \boldsymbol{v}_{j',t+1}) - \boldsymbol{v}_{1,t} + \boldsymbol{v}_{j',t}. \tag{13}$$

Interestingly, (12) is a separable optimization with respect to the coordinates of $\boldsymbol{\lambda}_1$, i.e., for each $i \in [n]$ we have

$$(\boldsymbol{\lambda}_{1,t+1})_i = \underset{\lambda \in \mathbb{R}}{\arg\min} \frac{1}{n}\ell_i\left(\frac{n}{\sigma}\left(\boldsymbol{\lambda}_{1,t+1/2}\right)_i - \lambda\right) + \frac{1}{2\sigma}\lambda^2.$$

Therefore, (12) admits efficient and parallelizable solvers.

## 3 Convergence guarantees

We begin by stating a few assumptions that will be used to provide convergence guarantees for the primal iterates $(\boldsymbol{\theta}_{j,t})_{t\geq 1}$. Recall the assumptions that the loss function $\ell(\cdot,\cdot)$ is *nonnegative* and the regularizer is separable as in (2). We will provide convergence rates for two different classes of loss functions. First, the Lipschitz loss functions, for which there exists a constant $\rho \geq 0$ such that

$$|\ell(u,w) - \ell(v,w)| \leq \rho|u-v|, \qquad\qquad \text{for all } u,v,w \in \mathbb{R}.$$

By differentiability of $\ell(\cdot,\cdot)$ in its first argument, the condition above is equivalent to

$$\left|\frac{\mathrm{d}\ell(u,v)}{\mathrm{d}u}\right| \leq \rho, \qquad\qquad \text{for all } u,v \in \mathbb{R}. \tag{Lip.}$$

Examples of the Lipschitz loss functions are the absolute loss, the Huber loss, and the logistic loss. Second, the *square root Lipschitz* loss functions, for which there exists a constant $\rho \geq 0$ such that

$$|\sqrt{\ell(u,w)} - \sqrt{\ell(v,w)}| \leq \frac{\rho}{2}|u-v|, \qquad\qquad \text{for all } u,v,w \in \mathbb{R}.$$

Again, invoking differentiability of $\ell(\cdot,\cdot)$ we can equivalently write

$$\left|\frac{\mathrm{d}\ell(u,v)}{\mathrm{d}u}\right| \leq \rho\sqrt{\ell(u,v)}. \qquad\qquad (\sqrt{}\text{-Lip.})$$

Examples of the square root Lipschitz loss functions are the squared loss, the Huber loss.

Furthermore, we assume that for some known constant $R > 0$ the empirical risk minimizer $\widehat{\boldsymbol{\theta}}$ is bounded as

$$\left\|\widehat{\boldsymbol{\theta}}\right\|_2 \leq R. \tag{minimizer bound}$$

We also assume that the agents are provided with the constant $\chi$ that bounds the usual operator norm of the design matrix as

$$\|\boldsymbol{X}\| \leq \chi. \tag{design bound}$$

The constants $\delta > 0$ that bounds the spectral gap of the network as

$$\left\|\boldsymbol{L}^{\dagger}\right\| \leq \delta^{-1}, \tag{spectral gap}$$

with $\boldsymbol{M}^{\dagger}$ denoting the Moore-Penrose pseudoinverse of the matrix $\boldsymbol{M}$, as well as the constant $D > 0$ that bounds the operator norm of the Laplacian as

$$\|\boldsymbol{L}\| \leq D, \tag{Laplacian bound}$$

are also provided to the agents. Because $n\|\boldsymbol{K}\| \leq \max_{j \in [m]} \|\boldsymbol{X}_j\| + \|\boldsymbol{L} \otimes \boldsymbol{I}\| \leq \|\boldsymbol{X}\| + \|\boldsymbol{L}\|$, instead of assuming an additional bound for $\|\boldsymbol{K}\|$, we will use the bound $\|\boldsymbol{K}\| \leq (\chi + D)/n$.

**Theorem 1.** *Suppose that the $m$ agents are given the positive constants $R$, $\chi$, $\delta$ and $D$ that respectively satisfy* (minimizer bound), (design bound), (spectral gap), *and* (Laplacian bound), *so that they can choose $\sigma = m^{1/2}n^{3/2}\rho / \left((\chi + D)R\sqrt{1 + 2\chi^2/\delta^2}\right)$ and $\tau = n^2 / \left((\chi + D)^2\sigma\right)$. Denote the temporal average of the vectors $\boldsymbol{\theta}_{j,t}$ over the first $T \geq 1$ iterations by*

$$\overline{\boldsymbol{\theta}}_j = \frac{1}{T}\sum_{t=1}^{T} \boldsymbol{\theta}_{j,t}, \qquad\qquad for\ j \in [m], \tag{14}$$

*and let $\overline{\boldsymbol{\theta}} = \begin{bmatrix} \overline{\boldsymbol{\theta}}_1; & \cdots; & \overline{\boldsymbol{\theta}}_m \end{bmatrix}$. Under the Lipschitz loss model* (Lip.) *we have*

$$\frac{1}{n}\sum_{i=1}^{n} \ell_i(\overline{\boldsymbol{\theta}}) + r(\overline{\boldsymbol{\theta}}) \leq \frac{1}{n}\sum_{i=1}^{n} \ell_i(\widehat{\boldsymbol{\theta}}) + r(\widehat{\boldsymbol{\theta}}) + \frac{2(\chi + D)R\rho}{(n/m)^{1/2}T}\sqrt{1 + \frac{2\chi^2}{\delta^2}} \tag{15}$$

*Similarly, under the square root Lipschitz loss model ($\sqrt{\ }$-Lip.) and for $T \geq 2mn\rho^2/\sigma$ we have an "isomorphic convergence" given by*

$$\frac{1}{n}\sum_{i=1}^{n} \ell_i(\overline{\boldsymbol{\theta}}) + r(\overline{\boldsymbol{\theta}}) \leq \left(1 + \frac{2(\chi + D)R\rho}{m^{1/2}n^{3/2}T}\sqrt{1 + \frac{2\chi^2}{\delta^2}}\right) \times \left(\frac{1}{n}\sum_{j=1}^{n} \ell_i\left((\boldsymbol{X}\widehat{\boldsymbol{\theta}})_i\right) + r(\widehat{\boldsymbol{\theta}}) + \frac{(\chi + D)R\rho}{(n/m)^{1/2}T}\sqrt{1 + \frac{2\chi^2}{\delta^2}}\right). \tag{16}$$

The prescribed $\tau$ and $\sigma$ are "optimized" for the Lipschitz model. The well-tuned choice of $\tau$ and $\sigma$ under the square root Lipschitz model is slightly different and depends on the minimum value of the objective. For simplicity, we used the former in the theorem for both models.

For a better understanding of the convergence bounds (15) and (16), it is worth considering more interpretable approximations of the quantities $D$ and $\delta$. With $\Delta(G)$ denoting the maximum degree of the graph $G$, we have an elementary bound $\|\boldsymbol{L}\| \leq 2\Delta(G)$, so it suffices to choose $D \geq 2\Delta(G)$. Furthermore, for a connected graph, $\left\|\boldsymbol{L}^{\dagger}\right\|$ is reciprocal to the second smallest eigenvalue of $\boldsymbol{L}$, and we can invoke an inequality due to Mohar (1991, Theorem 2.3) that relates the spectral gap, diameter, and the maximum degree of a graph, we have $\left\|\boldsymbol{L}^{\dagger}\right\| \geq 2\left(\text{diam}(G) - 1 - \log(m-1)\right)/\Delta(G)$ which can provide a general bound on how large $\delta$ can possibly be. Another inequality (Mohar, 1991, Theorem 4.2), attributed to Brendan Mckay, also provides the bound $\left\|\boldsymbol{L}^{\dagger}\right\| \leq m\,\text{diam}(G)/4$ which implies a conservative choice of $\delta \leq 4/(m\,\text{diam}(G))$. The networks that

are *(spectral) expanders* are more favorable as they typically have larger spectral gap and smaller maximum degree simultaneously. For instance, for $k$-regular *Ramanujan graphs* we can choose $\delta = k - 2\sqrt{k-1}$ (Lubotzky et al., 1988; Mohar, 1992).

The algorithm can be generalized by assigning weights to the edges of the network and choosing $\boldsymbol{L}$ to be the Laplacian of the weighted network. The effect of using weighted edges on the algorithm is that the simple summation iterates of the neighboring agents in (10), (11), and (13) (thereby (12)), will become weighted summations. Using weighted edges allows us, in principle, to optimize bounds (15) and (16) by adjusting the edge weights.

We have shown that we can solve (1) in the feature-distributed setting and achieve a convergence rate of $O(1/T)$ under relatively simple assumptions. The iterations each agent has to solve is rather simple, including (12) thanks to its separability. However, there are a few limitations in the proposed framework that have to be considered. First, the agents cannot rely only on local information to choose $\tau$ and $\sigma$; in general they can obtain the required global information at the cost of extra communications. Second, the scope of the algorithm is limited by the fact that the loss function acts on linear predictors $\boldsymbol{x}_i^\mathsf{T}\boldsymbol{\theta}$. It is worth mentioning, however, that this limitation is basically necessary to stay in the realm of convex optimization; we are not aware of any widely used nonlinear predictor whose composition with standard loss functions is convex. Third, the considered saddle-point formulation incurs a significant communication and computation cost associated with the iterates $(\boldsymbol{\lambda}_{j,t})$ and $(\boldsymbol{v}_{j,t})$; it is not clear if this is inherent to the problem.

## 4 Numerical Experiments

We provide several numerical experiments to illustrate the behavior of the proposed algorithm with varying quantities of agents and communication graphs. In the case where computation is of greater cost than communication, we find that our algorithm can make use of parallelism to improve performance.

We solve the least squares problem

$$\underset{\boldsymbol{\theta}}{\text{minimize}} \; \frac{1}{2}\|\boldsymbol{X}\boldsymbol{\theta} - \boldsymbol{y}\|_2^2$$

for a synthetic dataset of $2^{14} = 16384$ samples and $2^{11} = 2048$ features so that $\boldsymbol{X}$ is a $16384 \times 2048$ matrix. To construct the synthetic dataset, the design matrix $\boldsymbol{X}$, the ground truth vector $\boldsymbol{\theta}_\star$, and the noise vector $\boldsymbol{e}$ are all populated by i.i.d. samples of the standard normal distribution. The corresponding noisy response vector $\boldsymbol{y}$ is then computed as $\boldsymbol{y} = \boldsymbol{X}\boldsymbol{\theta}_\star + \boldsymbol{e}$. In all experiments, the features are partitioned equally among the agents, i.e., each agent has access to exactly $d/m$ features.

We explore the following communication graph structures:

- *Complete Graph*: All agents are connected to all other agents.

- *Star Graph*: All agents are connected only to the first agent.

- *Erdős–Rényi Graph*: Each of the possible $\binom{m}{2}$ pairs of agents are connected with probability $p \in \{0.1, 0.5\}$ independent of the other connections. To avoid violating the connectivity requirement of the communication graph (with high probability), we only consider graphs of 8 or more agents in the case $p = 0.5$, and graphs of 32 or more agents in the case of $p = 0.1$.

- *2D Lattice Graph*: The agents are arranged in 2D space as a square lattice. Each agent is connected to its cardinal and diagonal neighbors. The first agent is located at one of the four center-most lattice points.

- *Random Geometric Graph*: Agents are assigned positions in the 2D unit square uniformly at random. A pair of agents are connected if the Euclidean distance between their positions is less than 0.3. Again, to avoid violating the connectivity requirement of the communication graph (with high probability), we only consider 32 agents or more.

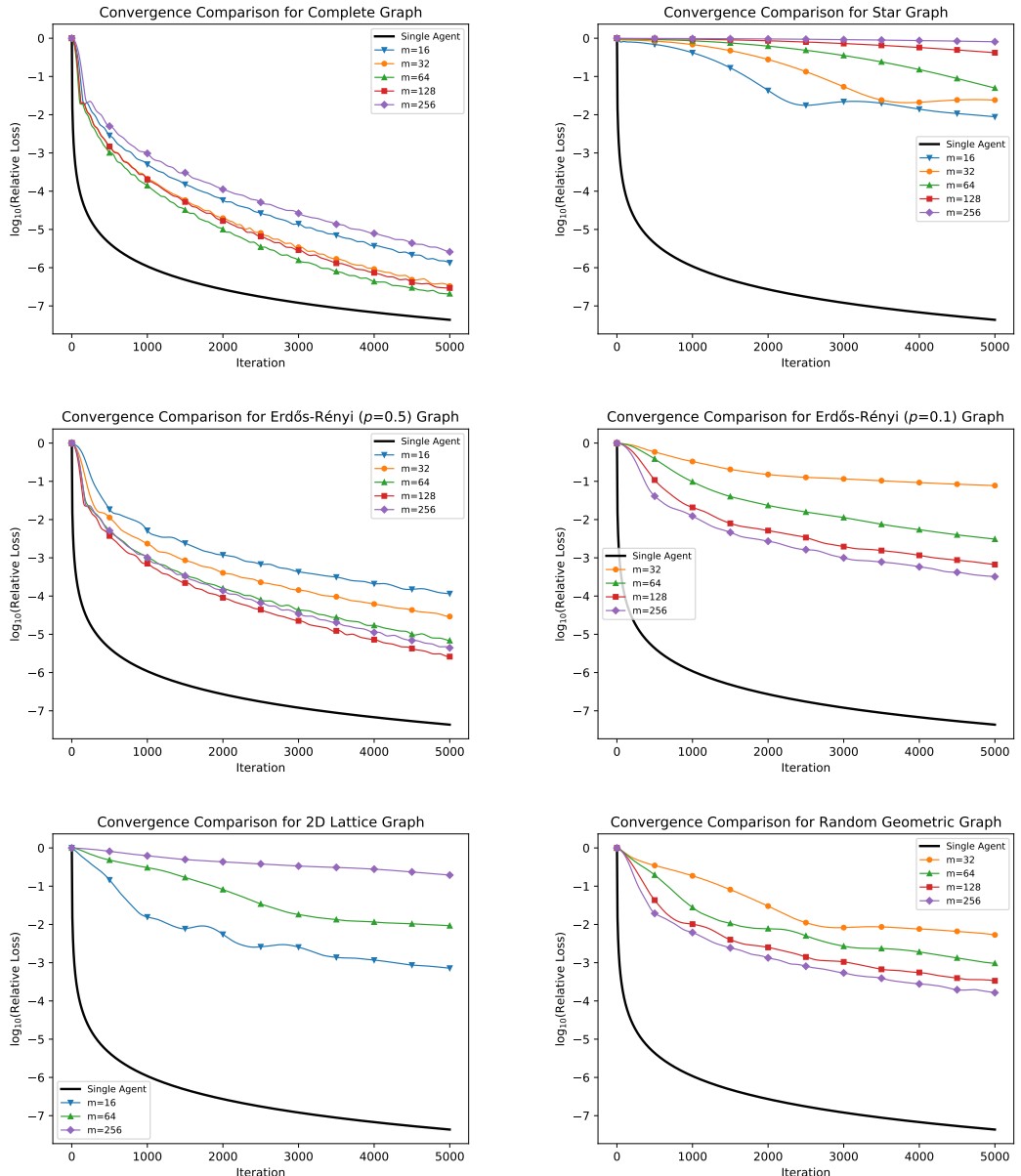

Figure 1: Plots depicting algorithm progress for varying communication graph structures and number of agents. The single agent progress is included in all plots for reference. With $\mathcal{L}_t$ denoting the objective (i.e., the regularized empirical risk) at $\bar{\boldsymbol{\theta}}_t$, and $\mathcal{L}_\star$ denoting the minimum value of the objective, the vertical axis represents the base-10 logarithm of the relative error defined as $\log_{10}\left(\frac{\mathcal{L}_t - \mathcal{L}_\star}{\mathcal{L}_0 - \mathcal{L}_\star}\right)$. The horizontal axis represents number of iterations completed.

As a baseline, we solve the single agent problem using the proposed primal-dual algorithm but with the Lagrange multiplier $\boldsymbol{v}$ terms fixed at zero, however we recognize that the problem choice could also be solved by other algorithms, e.g. gradient descent. (For the single agent case, the Laplacian constraints of (3) are trivially satisfied and can be omitted.) Figure 1 shows the convergence behavior of the proposed algorithm for each of the aforementioned communication graph structures. The complete graph tends to converge faster than any other graph for a fixed number of agents, and performs best at 64 agents (with 32 features per agent) instead of continually improving with increasing quantity of agents. Similarly, the Erdős-Rényi

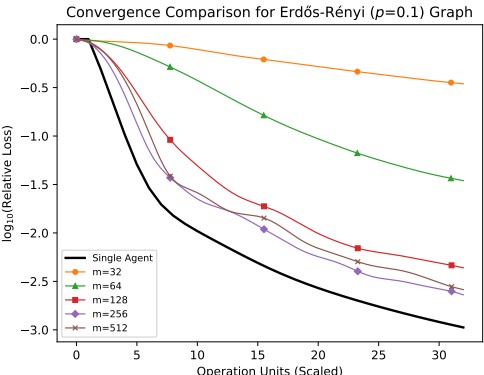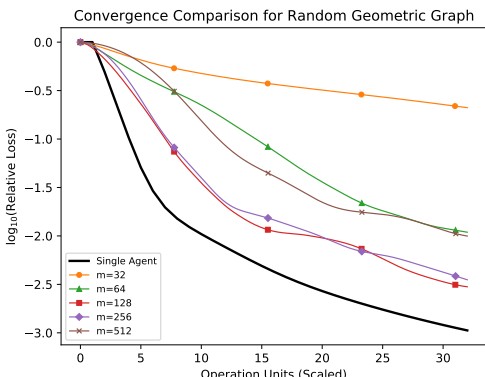

Figure 2: Plots depicting algorithm progress for Erdős-Rényi ($p = 0.1$) and random geometric graphs under the given cost paradigm. With $\mathcal{L}_t$ denoting the objective (i.e., the regularized empirical risk) at $\bar{\boldsymbol{\theta}}_t$, and $\mathcal{L}_\star$ denoting the minimum value of the objective, the vertical axis represents the base-10 logarithm of the relative error defined as $\log_{10}\left(\frac{\mathcal{L}_t - \mathcal{L}_\star}{\mathcal{L}_0 - \mathcal{L}_\star}\right)$. The horizontal axis represents units of operations per agent completed (not iteration) normalized such that the single agent case completes one iteration per unit of operation (i.e. the single agent completes 32 iterations). Explicitly, iteration $t$ corresponds to $\frac{n(4(d/m) + 2\Delta(G) + 7) + 5(d/m)}{n(4d+1) + 5d}t$ on the horizontal axis (except for the single agent case, where iteration $t$ corresponds to $t$ on the horizontal axis). In short, settings with fewer operations per agent per iteration complete more iterations.

graphs perform best at 128 and 256 agents for $p = 0.5$ and $p = 0.1$, respectively. Convergence degrades as $p$ decreases. The random geometric graph performs very similarly to the Erdős-Rényi graph for $p = 0.1$. Both the star and 2D lattice graphs perform increasingly worse as the quantity of agents increases. We speculate this is caused by a large quantity of comparatively small eigenvalues for the associated Laplacian matrices.

If we assume a situation where cost is dominated by computation rather than communication, the proposed algorithm can achieve comparable performance to the single agent case even under relatively sparse graphs. Recall that $n$, $m$, and $d$ represent the number of samples, agents, and features, respectively, and that $\Delta(G)$ denotes the maximum degree of the communication graph $G$. One can show that each iteration of the proposed algorithm requires each agent complete $n(4(d/m) + 2\Delta(G) + 7) + 5(d/m)$ floating point operations.[2] In the single agent case, one can show $n(4d + 1) + 5d$ floating point operations are needed per iteration.[3]

We also compare scenarios for a fixed number of operations per agent. As the number of agents increases $\boldsymbol{X}$ and $\boldsymbol{\theta}$ are increasingly split over more agents, effectively parallelizing the problem. This leads to a decrease in the number of operations per agent for the matrix-vector multiplies in (9) and (11) which dominate the operation cost. Figure 2 illustrates how, under this cost paradigm, the relatively sparse Erdős-Rényi ($p = 0.1$) and random geometric graphs with 256 agents achieve performance comparable to that of the single agent case. This speaks to the promise of the proposed algorithm for very large problem sizes over relatively sparse graphs.

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

# A  Proof of Theorem 1

As the dual parameters are not important for our purposes, our goal is to convert the established saddle-point convergence rates of the Chambolle–Pock algorithm (Chambolle & Pock, 2016) into primal convergence rates. Similar to (14), define the temporal average of the other iterates over the first $T$ iterations as

$$\overline{\boldsymbol{v}}_j = \frac{1}{T} \sum_{t=1}^{T} \boldsymbol{v}_{j,t}$$

$$\overline{\boldsymbol{\lambda}}_j = \frac{1}{T} \sum_{t=1}^{T} \boldsymbol{\lambda}_{j,t} \,,$$

for $j \in [m]$, and let $\overline{V} = \begin{bmatrix} \overline{v}_1 & \cdots & \overline{v}_m \end{bmatrix}$ and $\overline{\lambda} = \begin{bmatrix} \overline{\lambda}_1; & \cdots; & \overline{\lambda}_n \end{bmatrix}$. Furthermore, denote the objective of the saddle-point problem (4) by

$$\mathcal{E}(\boldsymbol{\theta}, \boldsymbol{V}, \boldsymbol{\lambda}) = \underbrace{\sum_{j=1}^{m} r_j(\boldsymbol{\theta}_j)}_{=r(\boldsymbol{\theta})} + \frac{1}{n}\boldsymbol{\lambda}_j^\top \left( \boldsymbol{X}_j\boldsymbol{\theta}_j + \boldsymbol{V}\boldsymbol{L}\boldsymbol{e}_j \right) - \frac{1}{n}\sum_{i=1}^{n} \ell_i^*(\lambda_{1,i}). \tag{17}$$

With the iterates initialized at zero (i.e., $\boldsymbol{\theta}_{j,0} = \boldsymbol{0}$, $\boldsymbol{v}_{j,0} = \boldsymbol{0}$, and $\boldsymbol{\lambda}_{j,0} = \boldsymbol{0}$ for all $j \in [m]$), and observing that

$$\tau\sigma\|\boldsymbol{K}\|^2 \leq 1\,,$$

we can apply the convergence rate established in (Chambolle & Pock, 2016, Theorem 1, and Remark 2) to obtain

$$\begin{aligned}
&\mathcal{E}(\overline{\boldsymbol{\theta}}, \overline{\boldsymbol{V}}, \boldsymbol{\lambda}) - \mathcal{E}(\boldsymbol{\theta}, \boldsymbol{V}, \overline{\boldsymbol{\lambda}}) \\
&\leq \frac{1}{T}\sum_{j=1}^{m}\left( \frac{1}{2\tau}\|\boldsymbol{\theta}_j - \boldsymbol{\theta}_{j,0}\|_2^2 + \frac{1}{2\tau}\|\boldsymbol{v}_j - \boldsymbol{v}_{j,0}\|_2^2 + \frac{1}{2\sigma}\|\boldsymbol{\lambda}_j - \boldsymbol{\lambda}_{j,0}\|_2^2 \right.\\
&\qquad\qquad \left. - \frac{1}{n}(\boldsymbol{\lambda}_j - \boldsymbol{\lambda}_{j,0})^\top \left( \boldsymbol{X}_j(\boldsymbol{\theta}_j - \boldsymbol{\theta}_{j,0}) + \sum_{j'\in[m]:\,j\sim_G j'} \boldsymbol{v}_j - \boldsymbol{v}_{j,0} - \boldsymbol{v}_{j'} + \boldsymbol{v}_{j',0} \right) \right) \\
&\leq \frac{1}{T}\sum_{j=1}^{m}\left( \frac{1}{\tau}\|\boldsymbol{\theta}_j - \boldsymbol{\theta}_{j,0}\|_2^2 + \frac{1}{\tau}\|\boldsymbol{v}_j - \boldsymbol{v}_{j,0}\|_2^2 + \frac{1}{\sigma}\|\boldsymbol{\lambda}_j - \boldsymbol{\lambda}_{j,0}\|_2^2 \right) \\
&= \frac{1}{T}\left( \frac{1}{\tau}\|\boldsymbol{\theta}\|_2^2 + \frac{1}{\tau}\|\boldsymbol{V}\|_{\mathrm{F}}^2 + \frac{1}{\sigma}\|\boldsymbol{\lambda}\|_2^2 \right),
\end{aligned}$$

for all $\boldsymbol{\theta}$, $\boldsymbol{V}$, and $\boldsymbol{\lambda}$. Rearranging the terms, we equivalently have

$$\mathcal{E}(\overline{\boldsymbol{\theta}}, \overline{\boldsymbol{V}}, \boldsymbol{\lambda}) - \frac{1}{T\sigma}\|\boldsymbol{\lambda}\|_2^2 \leq \mathcal{E}(\boldsymbol{\theta}, \boldsymbol{V}, \overline{\boldsymbol{\lambda}}) + \frac{1}{T\tau}\left( \|\boldsymbol{\theta}\|_2^2 + \|\boldsymbol{V}\|_{\mathrm{F}}^2 \right).$$

Recalling (17), taking the maximum of the left-hand side with respect to $\boldsymbol{\lambda}$, and applying Lemma 1 to the part corresponding to $\boldsymbol{\lambda}_1$, we have

$$\begin{aligned}
&r(\overline{\boldsymbol{\theta}}) + \frac{1}{n}\sum_{i=1}^{n}\ell_i\left( (\boldsymbol{X}_1\overline{\boldsymbol{\theta}}_1 + \overline{\boldsymbol{V}}\boldsymbol{L}\boldsymbol{e}_1)_i \right) - \frac{1}{T\sigma}\sum_{i=1}^{n}(\ell_i'((\boldsymbol{X}_1\overline{\boldsymbol{\theta}}_1 + \overline{\boldsymbol{V}}\boldsymbol{L}\boldsymbol{e}_1)_i))^2 + \sum_{j=2}^{m}\frac{T\sigma}{4n^2}\left\|\boldsymbol{X}_j\overline{\boldsymbol{\theta}}_j + \overline{\boldsymbol{V}}\boldsymbol{L}\boldsymbol{e}_j\right\|_2^2 \\
&\leq \min_{\boldsymbol{\theta}\in\mathbb{R}^d, \boldsymbol{V}\in\mathbb{R}^{n\times m}} r(\boldsymbol{\theta}) + \frac{1}{n}\sum_{j=1}^{m}\overline{\boldsymbol{\lambda}}_j^\top\left( \boldsymbol{X}_j\boldsymbol{\theta}_j + \boldsymbol{V}\boldsymbol{L}\boldsymbol{e}_j \right) - \frac{1}{n}\sum_{i=1}^{n}\ell_i^*(\overline{\lambda}_{1,i}) + \frac{1}{T\tau}\left( \|\boldsymbol{\theta}\|_2^2 + \|\boldsymbol{V}\|_{\mathrm{F}}^2 \right).
\end{aligned} \tag{18}$$

Next we establish a few more inequalities depending on the characteristics of the loss function, that together with (18) yield the desired convergence rates.

## A.1 Lower bound for the left-hand side of (18)

### A.1.1 Lipschitz loss

We first consider the case of Lipschitz loss functions (Lip.). Using convexity of $\ell_i(\cdot)$, we can write

$$\begin{aligned}
\frac{1}{n}\sum_{i=1}^{n}\ell_i\left( (\boldsymbol{X}_1\overline{\boldsymbol{\theta}}_1 + \overline{\boldsymbol{V}}\boldsymbol{L}\boldsymbol{e}_1)_i \right) &\geq \frac{1}{n}\sum_{i=1}^{n}\ell_i\left( (\boldsymbol{X}\overline{\boldsymbol{\theta}})_i \right) - \ell_i'((\boldsymbol{X}\overline{\boldsymbol{\theta}})_i)\sum_{j=2}^{m}\left( \boldsymbol{X}_j\overline{\boldsymbol{\theta}}_j + \overline{\boldsymbol{V}}\boldsymbol{L}\boldsymbol{e}_j \right)_i \\
&\geq \frac{1}{n}\sum_{i=1}^{n}\ell_i\left( (\boldsymbol{X}\overline{\boldsymbol{\theta}})_i \right) - \frac{m-1}{T\sigma}\sum_{i=1}^{n}(\ell_i'((\boldsymbol{X}\overline{\boldsymbol{\theta}})_i))^2 - \frac{T\sigma}{4n^2}\sum_{j=2}^{m}\left\|\boldsymbol{X}_j\overline{\boldsymbol{\theta}}_j + \overline{\boldsymbol{V}}\boldsymbol{L}\boldsymbol{e}_j\right\|_2^2,
\end{aligned}$$

where the second inequality is an application of the basic inequality $2ab \leq a^2 + b^2$. By construction, we have

$$\boldsymbol{X}_1\overline{\boldsymbol{\theta}}_1 + \overline{\boldsymbol{V}}\boldsymbol{L}\boldsymbol{e}_1 + \sum_{j=2}^m \boldsymbol{X}_j\overline{\boldsymbol{\theta}}_j + \overline{\boldsymbol{V}}\boldsymbol{L}\boldsymbol{e}_j = \boldsymbol{X}\overline{\boldsymbol{\theta}}\,.$$

Therefore, in view of (Lip.), what we have shown is

$$
\begin{aligned}
&r(\overline{\boldsymbol{\theta}}) + \frac{1}{n}\sum_{i=1}^n \ell_i\left((\boldsymbol{X}_1\overline{\boldsymbol{\theta}}_1 + \overline{\boldsymbol{V}}\boldsymbol{L}\boldsymbol{e}_1)_i\right) - \frac{1}{T\sigma}\sum_{i=1}^n (\ell_i'((\boldsymbol{X}_1\overline{\boldsymbol{\theta}}_1 + \overline{\boldsymbol{V}}\boldsymbol{L}\boldsymbol{e}_1)_i))^2 + \sum_{j=2}^m \frac{T\sigma}{4n^2}\big\|\boldsymbol{X}_j\overline{\boldsymbol{\theta}}_j + \overline{\boldsymbol{V}}\boldsymbol{L}\boldsymbol{e}_j\big\|_2^2 \\
&\geq \frac{1}{n}\sum_{i=1}^n \ell_i\left((\boldsymbol{X}\overline{\boldsymbol{\theta}})_i\right) + r(\overline{\boldsymbol{\theta}}) - \frac{mn\rho^2}{T\sigma}\,.
\end{aligned}
\tag{19}
$$

### A.1.2  Square root Lipschitz loss

The second case we consider is that of the square root Lipschitz loss functions ($\sqrt{\ }$-Lip.). It follows from ($\sqrt{\ }$-Lip.) that

$$\sum_{i=1}^n (\ell_i'((\boldsymbol{X}_1\overline{\boldsymbol{\theta}}_1 + \overline{\boldsymbol{V}}\boldsymbol{L}\boldsymbol{e}_1)_i))^2 \leq \rho^2 \sum_{i=1}^n \ell_i((\boldsymbol{X}_1\overline{\boldsymbol{\theta}}_1 + \overline{\boldsymbol{V}}\boldsymbol{L}\boldsymbol{e}_1)_i)\,.$$

For sufficiently large $T$ we have $\gamma \overset{\text{def}}{=} n\rho^2/(T\sigma) < 1/m$, and we can lower bound the left-hand side of (18), excluding the term $r(\overline{\boldsymbol{\theta}})$, as

$$
\begin{aligned}
&\frac{1}{n}\sum_{i=1}^n \ell_i\left((\boldsymbol{X}_1\overline{\boldsymbol{\theta}}_1 + \overline{\boldsymbol{V}}\boldsymbol{L}\boldsymbol{e}_1)_i\right) - \frac{1}{T\sigma}\sum_{i=1}^n (\ell_i'((\boldsymbol{X}_1\overline{\boldsymbol{\theta}}_1 + \overline{\boldsymbol{V}}\boldsymbol{L}\boldsymbol{e}_1)_i))^2 + \sum_{j=2}^m \frac{T\sigma}{4n^2}\big\|\boldsymbol{X}_j\overline{\boldsymbol{\theta}}_j + \overline{\boldsymbol{V}}\boldsymbol{L}\boldsymbol{e}_j\big\|_2^2 \\
&\geq (1-\gamma)\frac{1}{n}\sum_{i=1}^n \ell_i\left((\boldsymbol{X}_1\overline{\boldsymbol{\theta}}_1 + \overline{\boldsymbol{V}}\boldsymbol{L}\boldsymbol{e}_1)_i\right) + \sum_{j=2}^m \frac{T\sigma}{4n^2}\big\|\boldsymbol{X}_j\overline{\boldsymbol{\theta}}_j + \overline{\boldsymbol{V}}\boldsymbol{L}\boldsymbol{e}_j\big\|_2^2 \\
&\geq (1-\gamma)\left(\frac{1}{n}\sum_{i=1}^n \ell_i\left((\boldsymbol{X}\overline{\boldsymbol{\theta}})_i\right) - \ell_i'((\boldsymbol{X}\overline{\boldsymbol{\theta}})_i)\sum_{j=2}^m \left(\boldsymbol{X}_j\overline{\boldsymbol{\theta}}_j + \overline{\boldsymbol{V}}\boldsymbol{L}\boldsymbol{e}_j\right)_i\right) + \sum_{j=2}^m \frac{T\sigma}{4n^2}\big\|\boldsymbol{X}_j\overline{\boldsymbol{\theta}}_j + \overline{\boldsymbol{V}}\boldsymbol{L}\boldsymbol{e}_j\big\|_2^2,
\end{aligned}
\tag{20}
$$

where we used the convexity of the function $\ell_i(\cdot)$ in the second line. Again using the basic inequality $2ab \leq a^2 + b^2$, we have

$$
\begin{aligned}
&\frac{1}{n}\sum_{i=1}^n \ell_i'((\boldsymbol{X}\overline{\boldsymbol{\theta}})_i)\sum_{j=2}^m \left(\boldsymbol{X}_j\overline{\boldsymbol{\theta}}_j + \overline{\boldsymbol{V}}\boldsymbol{L}\boldsymbol{e}_j\right)_i \\
&\leq \sum_{i=1}^n \frac{(1-\gamma)(m-1)}{T\sigma}\left(\ell_i'((\boldsymbol{X}\overline{\boldsymbol{\theta}})_i)\right)^2 + \frac{T\sigma}{4(1-\gamma)n^2}\sum_{j=2}^m \left(\boldsymbol{X}_j\overline{\boldsymbol{\theta}}_j + \overline{\boldsymbol{V}}\boldsymbol{L}\boldsymbol{e}_j\right)_i^2 \\
&= \frac{(1-\gamma)(m-1)}{T\sigma}\sum_{i=1}^n \left(\ell_i'((\boldsymbol{X}\overline{\boldsymbol{\theta}})_i)\right)^2 + \frac{T\sigma}{4(1-\gamma)n^2}\sum_{j=2}^m \big\|\boldsymbol{X}_j\overline{\boldsymbol{\theta}}_j + \overline{\boldsymbol{V}}\boldsymbol{L}\boldsymbol{e}_j\big\|_2^2.
\end{aligned}
\tag{21}
$$

By ($\sqrt{\ }$-Lip.) we also have

$$\sum_{i=1}^n (\ell_i'((\boldsymbol{X}\overline{\boldsymbol{\theta}})_i))^2 \leq \rho^2 \sum_{i=1}^n \ell_i((\boldsymbol{X}\overline{\boldsymbol{\theta}})_i)\,,$$

which together with (20) and (21), and by adding back the term $r(\overline{\boldsymbol{\theta}})$, yields

$$r(\overline{\boldsymbol{\theta}}) + \frac{1}{n}\sum_{i=1}^n \ell_i\left((\boldsymbol{X}_1\overline{\boldsymbol{\theta}}_1 + \overline{\boldsymbol{V}}\boldsymbol{L}\boldsymbol{e}_1)_i\right) - \frac{1}{T\sigma}\sum_{i=1}^n (\ell_i'((\boldsymbol{X}_1\overline{\boldsymbol{\theta}}_1 + \overline{\boldsymbol{V}}\boldsymbol{L}\boldsymbol{e}_1)_i))^2$$

$$+ \sum_{j=2}^{m} \frac{T\sigma}{4n^2} \left\| \boldsymbol{X}_j \overline{\boldsymbol{\theta}}_j + \overline{\boldsymbol{V}} \boldsymbol{L} \boldsymbol{e}_j \right\|_2^2$$

$$\geq r(\overline{\boldsymbol{\theta}}) + (1 - \gamma)\left(1 - \gamma(1-\gamma)(m-1)\right) \frac{1}{n} \sum_{i=1}^{n} \ell_i \left((\boldsymbol{X}\overline{\boldsymbol{\theta}})_i\right)$$

$$\geq (1 - m\gamma) \left( \frac{1}{n} \sum_{i=1}^{n} \ell_i \left((\boldsymbol{X}\overline{\boldsymbol{\theta}})_i\right) + r(\overline{\boldsymbol{\theta}}) \right) \tag{22}$$

### A.2 Upper bound for the right-hand side of (18)

Furthermore, the right-hand side of the inequality (18) can be bounded as

$$\min_{\boldsymbol{\theta} \in \mathbb{R}^d, \boldsymbol{V} \in \mathbb{R}^{n \times m}} r(\boldsymbol{\theta}) + \frac{1}{n} \sum_{j=1}^{m} \overline{\boldsymbol{\lambda}}_j^{\mathsf{T}} \left(\boldsymbol{X}_j \boldsymbol{\theta}_j + \boldsymbol{V} \boldsymbol{L} \boldsymbol{e}_j\right) - \frac{1}{n} \sum_{i=1}^{n} \ell_i^*(\overline{\boldsymbol{\lambda}}_{1,i}) + \frac{1}{T\tau} \left( \|\boldsymbol{\theta}\|_2^2 + \|\boldsymbol{V}\|_{\mathrm{F}}^2 \right)$$

$$\leq \min_{\boldsymbol{\theta} \in \mathbb{R}^d, \boldsymbol{V} \in \mathbb{R}^{n \times m}} r(\boldsymbol{\theta}) + \frac{1}{n} \sum_{i=1}^{n} \ell_i \left((\boldsymbol{X}_1 \boldsymbol{\theta}_1 + \boldsymbol{V} \boldsymbol{L} \boldsymbol{e}_1)_i\right) + \frac{1}{n} \sum_{j=2}^{m} \overline{\boldsymbol{\lambda}}_j^{\mathsf{T}} \left(\boldsymbol{X}_j \boldsymbol{\theta}_j + \boldsymbol{V} \boldsymbol{L} \boldsymbol{e}_j\right) + \frac{1}{T\tau} \left( \|\boldsymbol{\theta}\|_2^2 + \|\boldsymbol{V}\|_{\mathrm{F}}^2 \right).$$

Imposing the constraints $\boldsymbol{X}_j \boldsymbol{\theta}_j + \boldsymbol{V} \boldsymbol{L} \boldsymbol{e}_j = \boldsymbol{0}$ for $j = 2, \ldots, m$, can only increase the value of minimum on the right-hand side. Namely, we have

$$\min_{\boldsymbol{\theta} \in \mathbb{R}^d, \boldsymbol{V} \in \mathbb{R}^{n \times m}} r(\boldsymbol{\theta}) + \frac{1}{n} \sum_{j=1}^{m} \overline{\boldsymbol{\lambda}}_j^{\mathsf{T}} \left(\boldsymbol{X}_j \boldsymbol{\theta}_j + \boldsymbol{V} \boldsymbol{L} \boldsymbol{e}_j\right) - \frac{1}{n} \sum_{i=1}^{n} \ell_i^*(\overline{\boldsymbol{\lambda}}_{1,i}) + \frac{1}{T\tau} \left( \|\boldsymbol{\theta}\|_2^2 + \|\boldsymbol{V}\|_{\mathrm{F}}^2 \right)$$

$$\leq \min_{\boldsymbol{\theta} \in \mathbb{R}^d, \boldsymbol{V} \in \mathbb{R}^{n \times m}} r(\boldsymbol{\theta}) + \frac{1}{n} \sum_{i=1}^{n} \ell_i \left((\boldsymbol{X}_1 \boldsymbol{\theta}_1 + \boldsymbol{V} \boldsymbol{L} \boldsymbol{e}_1)_i\right) + \frac{1}{T\tau} \left( \|\boldsymbol{\theta}\|_2^2 + \|\boldsymbol{V}\|_{\mathrm{F}}^2 \right)$$

$$\text{subject to } \boldsymbol{X}_j \boldsymbol{\theta}_j + \boldsymbol{V} \boldsymbol{L} \boldsymbol{e}_j = \boldsymbol{0}, \text{ for } j \in [m] \backslash \{1\}$$

$$\leq \min_{\boldsymbol{V} \in \mathbb{R}^{n \times m}} \frac{1}{T\tau} \|\boldsymbol{V}\|_{\mathrm{F}}^2 + \frac{1}{n} \sum_{i=1}^{n} \ell_i \left((\boldsymbol{X}\widehat{\boldsymbol{\theta}})_i\right) + r(\widehat{\boldsymbol{\theta}}) + \frac{1}{T\tau} \left\|\widehat{\boldsymbol{\theta}}\right\|_2^2$$

$$\text{subject to } \boldsymbol{X}_j \widehat{\boldsymbol{\theta}}_j + \boldsymbol{V} \boldsymbol{L} \boldsymbol{e}_j = \boldsymbol{0}, \text{ for } j \in [m] \backslash \{1\}$$

$$\leq \frac{1}{n} \sum_{i=1}^{n} \ell_i \left((\boldsymbol{X}\widehat{\boldsymbol{\theta}})_i\right) + r(\widehat{\boldsymbol{\theta}}) + \frac{1}{T\tau} \left\|\widehat{\boldsymbol{\theta}}\right\|_2^2 + \frac{1}{T\tau} \left\|(\boldsymbol{L} \otimes \boldsymbol{I})^{\dagger}\right\|^2 \left\| \left[\textstyle\sum_{j=2}^{m} \boldsymbol{X}_j \widehat{\boldsymbol{\theta}}_j; \quad -\boldsymbol{X}_2 \widehat{\boldsymbol{\theta}}_2; \quad \cdots; \quad -\boldsymbol{X}_m \widehat{\boldsymbol{\theta}}_m \right] \right\|_2^2$$

$$\leq \frac{1}{n} \sum_{i=1}^{n} \ell_i \left((\boldsymbol{X}\widehat{\boldsymbol{\theta}})_i\right) + r(\widehat{\boldsymbol{\theta}}) + \frac{1}{T\tau} \left\|\widehat{\boldsymbol{\theta}}\right\|_2^2 + \frac{2}{T\tau} \left\|(\boldsymbol{L} \otimes \boldsymbol{I})^{\dagger}\right\|^2 \|\boldsymbol{X}\|^2 \left\|\widehat{\boldsymbol{\theta}}\right\|_2^2, \tag{23}$$

where $\widehat{\boldsymbol{\theta}}$ is the empirical risk minimizer given by (1), and we used the bound

$$\left\| \left[\textstyle\sum_{j=2}^{m} \boldsymbol{X}_j \widehat{\boldsymbol{\theta}}_j; \quad -\boldsymbol{X}_2 \widehat{\boldsymbol{\theta}}_2; \quad \cdots; \quad -\boldsymbol{X}_m \widehat{\boldsymbol{\theta}}_m \right] \right\|_2^2 \leq \|\boldsymbol{X}\|^2 \left\|\widehat{\boldsymbol{\theta}}\right\|_2^2 + \max_{j \in [m] \backslash \{1\}} \|\boldsymbol{X}_j\|^2 \left\|\widehat{\boldsymbol{\theta}}\right\|_2^2.$$

### A.3 Convergence of the regularized empirical risk

We are now ready to derive the convergence rates under the loss models (Lip.) and ($\sqrt{\ }$-Lip.).

#### A.3.1 Lipschitz loss

In the case of Lipschitz loss model (Lip.), the bounds (18), (19), and (23) guarantee that

$$\frac{1}{n} \sum_{i=1}^{n} \ell_i \left((\boldsymbol{X}\overline{\boldsymbol{\theta}})_i\right) + r(\overline{\boldsymbol{\theta}}) \leq \frac{1}{n} \sum_{i=1}^{n} \ell_i \left((\boldsymbol{X}\widehat{\boldsymbol{\theta}})_i\right) + r(\widehat{\boldsymbol{\theta}}) + \frac{1}{T\tau} \left(1 + 2\left\|(\boldsymbol{L} \otimes \boldsymbol{I})^{\dagger}\right\|^2 \|\boldsymbol{X}\|^2\right) \left\|\widehat{\boldsymbol{\theta}}\right\|_2^2 + \frac{mn\rho^2}{T\sigma}$$

$$\leq \frac{1}{n}\sum_{i=1}^{n}\ell_i\left((\boldsymbol{X}\widehat{\boldsymbol{\theta}})_i\right) + r(\widehat{\boldsymbol{\theta}}) + \frac{1}{T\tau}\left(1 + \frac{2\chi^2}{\delta^2}\right)R^2 + \frac{mn\rho^2}{T\sigma}\,.$$

Using the values of $\sigma$ and $\tau$ prescribed by Theorem 1, we get (15).

### A.3.2 Square root Lipschitz loss

Similarly, for square root Lipschitz losses ($\sqrt{\phantom{x}}$-Lip.), it follows from (18), (22), and (23) that for $T \geq 2mn\rho^2/\sigma$ we have

$$\frac{1}{n}\sum_{i=1}^{n}\ell_i\left((\boldsymbol{X}\overline{\boldsymbol{\theta}})_i\right) \leq \left(1 - \frac{mn\rho^2}{T\sigma}\right)^{-1}\left(\frac{1}{n}\sum_{i=1}^{n}\ell_i\left((\boldsymbol{X}\widehat{\boldsymbol{\theta}})_i\right) + r(\widehat{\boldsymbol{\theta}}) + \frac{1}{T\tau}\left(1 + 2\left\|(\boldsymbol{L}\otimes\boldsymbol{I})^\dagger\right\|^2\|\boldsymbol{X}\|^2\right)\left\|\widehat{\boldsymbol{\theta}}\right\|_2^2\right)$$

$$\leq \left(1 + \frac{2mn\rho^2}{T\sigma}\right)\left(\frac{1}{n}\sum_{i=1}^{n}\ell_i\left((\boldsymbol{X}\widehat{\boldsymbol{\theta}})_i\right) + r(\widehat{\boldsymbol{\theta}}) + \frac{1}{T\tau}\left(1 + \frac{2\chi^2}{\delta^2}\right)R^2\right)\,.$$

Using the values of $\sigma$ and $\tau$ prescribed by Theorem 1 yields (16).

### A.4 Auxiliary Lemma

**Lemma 1.** *Let $f_1$ and $f_2$ be differentiable (closed) convex functions defined over a linear space $\mathcal{X}$. Denote their corresponding convex conjugate functions defined on the dual space $\mathcal{X}^*$ respectively by $f_1^*$ and $f_2^*$. For all $\boldsymbol{u} \in \mathcal{X}$ we have*

$$\max_{\boldsymbol{v}\in\mathcal{X}^*}\langle\boldsymbol{u},\boldsymbol{v}\rangle - (f_1^*(\boldsymbol{v}) + f_2^*(\boldsymbol{v})) \geq f_1(\boldsymbol{u}) - f_2^*(\nabla f_1(\boldsymbol{u}))\,.$$

*Proof.* The result follows from the duality of summation and *infimal convolution* (Bauschke & Combettes, 2011, Proposition 13.24), that is

$$\max_{\boldsymbol{v}\in\mathcal{X}^*}\langle\boldsymbol{u},\boldsymbol{v}\rangle - (f_1^*(\boldsymbol{v}) + f_2^*(\boldsymbol{v})) = \min_{\boldsymbol{w}\in\mathcal{X}}f_1(\boldsymbol{u}-\boldsymbol{w}) + f_2(\boldsymbol{w})$$

$$\geq \min_{\boldsymbol{w}\in\mathcal{X}}f_1(\boldsymbol{u}) - \langle\nabla f_1(\boldsymbol{u}),\boldsymbol{w}\rangle + f_2(\boldsymbol{w})$$

$$= f_1(\boldsymbol{u}) - f_2^*(\nabla f_1(\boldsymbol{u}))\,.$$

$\square$

## B  Supplementary Code and Figures

All code may be found in the supplementary materials file, along with a read-me file which details how to reproduce the results.

In an attempt to mimic the results of Figure 1, 101 trials were run for each setting. The random number generator was given a unique seed in each trial to generate different values. As a result, all of the problem tensors ($\boldsymbol{X}, \boldsymbol{\theta}_\star, \boldsymbol{e}$ and by consequence $\boldsymbol{y}$) along with graph structure (for random graphs) were re-sampled in each trial. Figure 3 shows the statistic results from these trials. It appears that convergence is most influenced by choice of graph structure as the random graphs tend to have a higher variance. However, intuitively this influence diminishes as the number of agents increases. A non-random graph structure yields nearly no perceivable spread, implying that the problem description is sufficiently robust for our purposes.

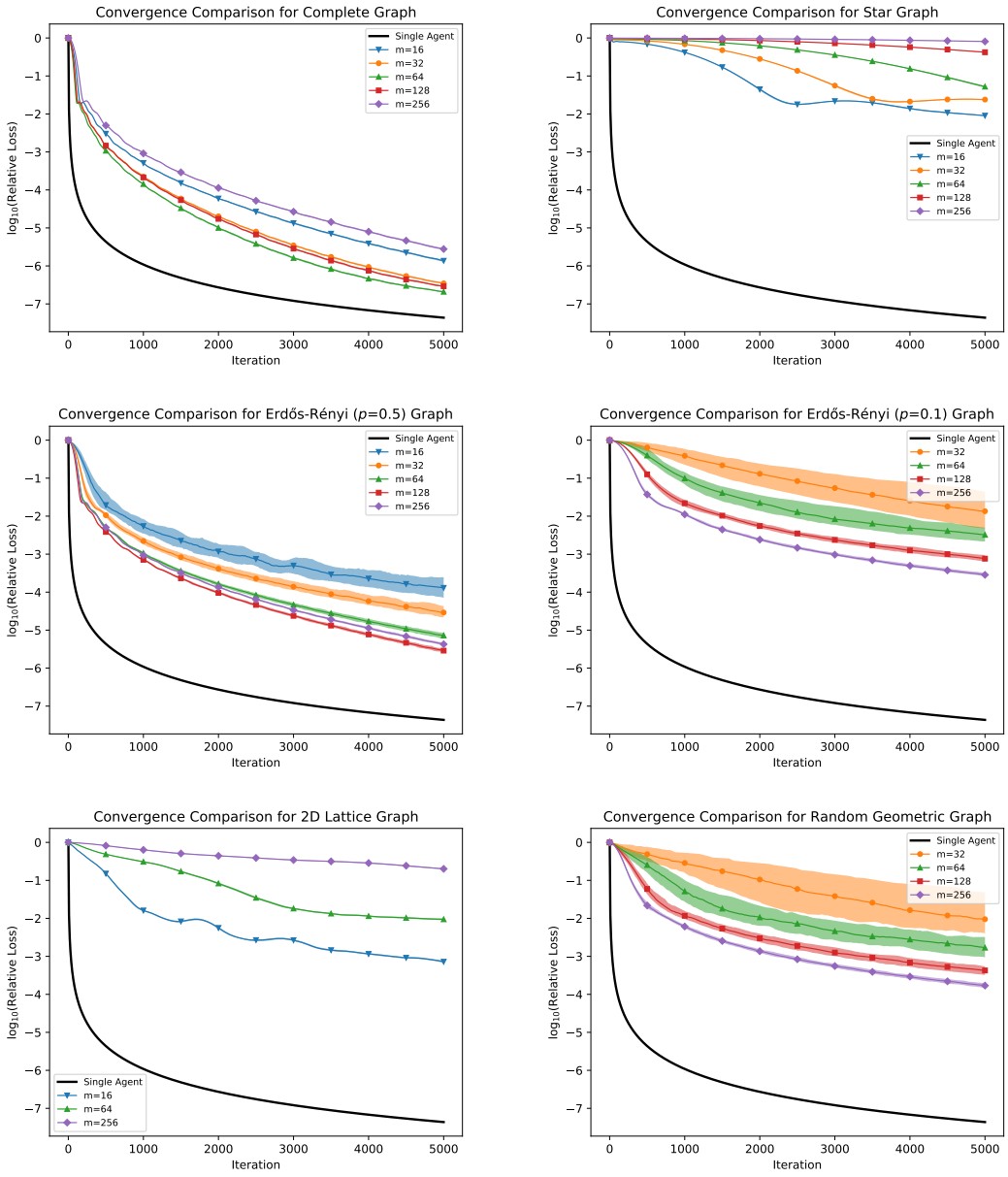

Figure 3: Plots highlighting the effect of randomness on convergence results. All plots are analogous to those in Figure 1, however the series' values now portray the median loss while the shaded regions portray the $10^{\text{th}}$ to $90^{\text{th}}$ percentile over 101 trials iteration-wise. For the results of non-random graphs (i.e. the single agent, complete graph, star graph, and 2D lattice graph) the difference between $10^{\text{th}}$ and $90^{\text{th}}$ percentile is not readily visible, but has indeed been plotted.

