# OpenReview forum: "Decentralized Feature-Distributed Optimization for Generalized Linear Models"
_TMLR — Rejected by TMLR_

### Review · Reviewer_uWxg · 2022-09-25

**Summary Of Contributions:**

This work proposes to solve the problem of distributed minimization with linear predictors and a regularizer, where the features have been partitioned across the nodes (agents). The authors rewrite the problem as a prima-dual one using convex conjugates of the empirical losses. On top of that, the authors introduce a consensus constraint to make it solvable in a distributed fashion; for the dualized problem with the constraints, they the Lagrangian as the (minmax) objective and apply Chambolle-Pock algorithm to it. The authors then study the proposed algorithms numerically on the problem of quadratic minimization.

I found the overall contribution to be too small. The work does not offer a substantial theoretical study, does not have significant algorithmic contributions and the numerical studies were done only to validate the theory. Therefore, I suggest rejecting this paper.

**Broader Impact Concerns:**

This work is mostly theoretical and is unlikely to have any ethical implications.

**Requested Changes:**

I do not expect this work to pass the acceptance threshold and, thus, do not request any significant changes. I did find a couple of typos  and one missing reference, listed below:
Equation (1) should end with a comma rather than period.
Page 4, "square root Lipschitz" -> "square-root Lipschitz"
Missing reference. The authors claim to consider "non-standard distributed setting where the data features, rather than samples, are distributed among m agents". This is, however, not that unusual as it has been considered for instance by Richtarik & Takac (2016) "Distributed Coordinate Descent Method for Learning with Big Data". Citing this reference would put the work into context.

**Strengths And Weaknesses:**

## Strengths
**Writing**. I found the writing easy to follow and contain almost no typos or mistakes. The presentation is clear, and the related work seems to be covered.

**Correctness**. I did not find any correctness issues.

## Weaknesses
**Motivation**. The paper lacks motivation. Parts missing motivation:
1. Why is the problem where the features are partitioned of interest? Is it motivated by a practical scenario where each agent has access to a subset of features or is the goal to make distributed optimization on a cluster run faster?
2. "We assume that only one of the agents, say the first agent, observes the response" I wish this assumption was explained better and justified.
3. What is the purpose of formulating the problem as a decentralized one? Is it to link it to a specific application or just to make the problem more general?

**Lack of novelty**. There is hardly much new in how the problem was dualized and how the Lagrangian was used to make it distributed. In particular, dualization is standard in the literature on primal-dual methods, while the feature partitioning has been previously considered in the work of Gratton et al. (2018), who used a similar approach in the context of quadratic problems and used an ADMM-based solver for it.

**Questionable practicality**. The methods presented here are somewhat old-fashioned and rely on evaluations of proximal operators instead of gradients. There are, of course, applications, where proximal operators can be computed efficiently, but it is still a limitation as gradient oracle is available more commonly.

**Limited numerical comparison**. The experiments do not compare the method to any other in the literature and do not make the method's motivation more apparent.

---

> ### Author Response · Authors · 2022-10-20
> **response to the comments of Reviewer uWxg**
>
> We thank the reviewer for helpful comments, below we address those listed under Weaknesses and the point about a reference mentioned under Requested changes.
>
> Motivation:
> > 1. & 3. We have already mentioned in the second paragraph of Section 1.1 the motivating examples such as sensor fusion, federated learning, and parallel computing where it makes sense to have data split across feature. To have a concrete example, one can imagine a swarm of small robots that have limited computational and communication budget that must collaboratively solve a classification problem based on their collective data. Variable selection using LASSO in any application domain with very high-dimensional data is another example where splitting across features is reasonable.
> 2. Each agent can have its own response variable, but the problems that the agents have to solve are similar. Therefore, it suffices to focus on the problem that one agent needs to solve.
>
> Lack of novelty:
> > We have already explained in Section 1.1 the distinctions from prior work. The work of Gratton et al. (2018) is heavily dependent on the structure of the least-squares problem and only provides asymptotic convergence guarantees.
>
> Questionable practicality:
> > As mentioned at the end of Section 2, the minimization in (12) decomposes to minimization of univariate strongly convex functions and, as such, they can be computed efficiently.  The proximal operators in (8) correspond to the regularizer in the GLM and their appearance is standard in non-smooth convex optimization.
>
> Limited numerical comparison:
> > The purpose of the experiments was to illustrate the role of graph structure in the convergence. It is hard to draw comparisons against COLA, as the per-iteration cost is so different.
>
> Paper by Richtarik & Takac [2016]:
> > The paper by Richtarik and Takac (2016) is certainly relevant and was missed merely because of nonoverlapping keywords; we will discuss this paper in the revised manuscript. Here we would like to mention that [RT16] focuses on the smooth and strongly convex setting and argues that the result can be extended even if the strong convexity is relaxed to convexity, but does not consider convex and Lipschitz objectives.

---

### Review · Reviewer_dA5i · 2022-09-30

**Summary Of Contributions:**

The paper studies the decentralized feature-distributed optimizaiton for generalized linear models.

The contribution can be summarized below:
* Providing a tight theoretical analysis for the case of feature-distributed optimization for decentralized learning is interesting and crucial.
* The paper applies the Chambolle-Pock primal-dual algorithm to reformulate the decentralized feature-distributed optimization problem, so as to give rates to the broader class of loss functions.

**Requested Changes:**

Please check the four weak points mentioned above.

**Strengths And Weaknesses:**

# Strengths
* The manuscript in general is well-written and well-structured.
* The studied problem is interesting and crucial to the decentralized optimization community.

# Weaknesses
The main issues of the current manuscript are the novelty and significance, due to the
1. Incomplete related work. The considered feature-distributed decentralized optimization has been extensively studied by the federated learning community, a special variant of decentralized learning with the star topology, i.e., vertical federated learning. However, the latest relevant work mentioned in the current manuscript was Chen et al., 2020. There should exist a large volume of Vertical Federated Learning (VFL) research and the manuscript needs to discuss them.
2. The authors need to compare their theoretical results (by modifying the L to match the star topology) with existing rates developed in Vertical Federated Learning (VFL).
3. The numerical results omit the comparison with other methods, e.g., COLA, and other VFL methods (for star topology). As the manuscript argues that its analysis can be applied to a broader class of square root Lipschitz loss functions, it would be great if the authors can construct some synthetic cases to empirically verify this point, so as to highlight the tightness of the proposed analysis.
4. The reviewer is also uncertain about (did not carefully check) the technical difficulty of extending the Chambolle-Pock primal-dual algorithm to the considered feature-distributed decentralized learning scenario. Authors are required to justify this point.

---

> ### Author Response · Authors · 2022-10-20
> **response to the comments of Reviewer dA5i**
>
> We thank the reviewer for useful comments; below we address the comments listed under weaknesses.
> > 1 & 2. We will cite the VFL paper in the revised manuscript.
>
> > 3. Let us clarify that our statement is "... our convergence analysis applies to **the** broader class of square root Lipschitz loss functions ...", by which we mean that the set of square root Lipschitz functions includes the set of (nonnegative) smooth functions as a subset. We can compare the performance against the VFL, but the VFL applies to a very particular network topology; of course if you specialize to a fixed topology, you can specialize the algorithm as well.

---

### Review · Reviewer_TYg7 · 2022-10-07

**Summary Of Contributions:**

This paper studies regularized ERM for generalized linear models where both the loss functions and the regularizer are convex. Further, the paper considers the decentralized setting specified using a time-invariant communication graph, where each machine has access to a disjoint set of features for all data points, and the regularize is separable across these features. Among other scenarios, this distributed setting arises when collecting data through multiple sensors, but it isn't very well understood.

The authors reduce the original undistributed optimization problem into a distributed saddle point problem and then solve it using a known primal-dual algorithm given appropriate proximal accesses. And provide convergence guarantees for Lipschitz and "square-root Lipschitz" loss functions. Numerical experiments study the effect of varying the communication graphs and the number of machines for the proposed algorithm.

**Broader Impact Concerns:**

There are no apparent ethical concerns with the submission.

**Requested Changes:**

1. Authors should remark on which common regularizers are separable and which are not.

2. I agree that the problem in (3) is amenable to decentralized distributed optimization and is more complicated than (1), so solving it should give a guarantee for (1). But the paper doesn't discuss why/how the Laplacian constraint in (3) helps their analysis.

3. I believe [this](https://arxiv.org/abs/2003.10422) is a missing reference for decentralized first-order methods.

4. The paper discussed federated learning literature but didn't mention *vertical federated learning* which considers a similar distributed features problem. [This](https://arxiv.org/abs/2202.04309) is a representative paper.

5. I like the flow of section 2, showing how to implement the Chambolle-Pock algorithm in their setting. However, it would be helpful to summarize all the machines' updates in closed form or through a prox-oracle in pseudocode. This code should also emphasize when communication happens between the nodes. I would further recommend defining the prox oracles in section two and using specific access to them in the pseudocode. This presentation will make the computational and communication complexity very clear. Finally, the pseudocode should highlight the critical point that (12) can be parallelized. Please clarify if this happens on new machines or the machines in the communication cluster of machine one.

6. Given the direct access to the prox oracles, discussing when such access can be efficiently implemented in practice is essential. This can be achieved by considering several example functions in section 2, either through remarks or a separate subsection. Mentioning when such oracles are infeasible to implement is also important to give the unfamiliar reader a balanced view of the contribution. Extending the work to inexact prox-oracles can be an excellent direction to add technical novelty.

7. Instead of tuning the hyper-parameters sub-optimally for two different loss classes, consider splitting the theorem into separate theorems.

8. It is good that the authors discuss some natural bounds on their problem parameters below the theorem. However, it would be better if corollaries were provided below the theorem to re-state the theorem in terms of these fundamental properties of $G$. It will also make it easier to interpret the provided guarantee.

9. Several different communication graphs are considered in the experiments. Do the theoretical guarantees for those graphs, which can be specialized by estimating their different bounds, predict the actual performance well? The authors should consider adding corollaries for all these special cases, at least in the appendix. The overall aim of this exercise is to expose the theoretical contribution well. As of now, it is unclear what to make of the different experiments. It would also be helpful to remark if these communication graphs capture real-world behavior.


**Strengths And Weaknesses:**

**Strength:** The paper studies an important understudied problem with broad applications. I like the general flow of the paper; the authors are clear about reducing the original non-distributed problem to a distributed one and proceeding from there on.

**Weakness:** I believe the authors can add several clarifying remarks to the paper to improve the writing. In particular, the final algorithm is not exposed very well, making it difficult to understand the total number and nature of the oracle queries required. The final convergence results can be described more simply in settings with natural scaling making the contribution clear. Some relevant papers are also missing in the related works section. Please see the next section for more details.

**Verdict:** This paper studies a critical problem that has not been studied extensively. The main contribution is casting the problem into a tractable saddle point problem which can be solved using a known algorithm given appropriate proximal accesses. The algorithmic insight in the paper is limited to showing how to implement the Chambolle-Pock algorithm in their setting. Further, I believe the authors can improve their writing in several places. I recommend revising the paper to alleviate these issues (see next section) and adding more results. I do not recommend accepting the paper's current form, but my review is borderline and might improve with a revised version.

---

> ### Author Response · Authors · 2022-10-20
> **response to the comments of Reviewer TYg7**
>
> We thank the reviewer for helpful comments; below we address the points raised by the reviewer.
>
> Weaknesses:
> > We will expand on the discussions in Section 2 in the revised manuscript to provide a better exposition of the algorithm.  Let us emphasize that without a regularizer, the algorithm amounts to each node performing a closed-form update and the main node solving n decoupled univariate strongly convex minimization problems. If there is a regularizer, then the nodes also have to evaluate the relevant prox operators corresponding to the regularizer.
>
> Requested changes:
> - separable regularizers
> > The most common separable regularizers include the $p$-th power of the $\ell_p$ norms as well as the mixed norms (e.g., $\ell_1$ norm of $\ell_2$ norms) defined by non-overlapping partitions of the coordinates.
> - Laplacian constraint in (3)
> > In decentralized optimization it is standard that the Laplacian of the communication network enters the formulation as an effective "regularizer" to enforce the "concensus" among the local copies of the optimization variables between neighboring nodes of the network. Basically, the regularization/constraint imposed based on the Laplacian, ensures the local copies of the optimization variables are all equal to a corresponding global variable.
> - First-order method reference
> >  We will add a discussion on this paper among the sample-distributed methods.
> - Vertical Federated Learning paper
> > We will cite the VFL paper in the revised manuscript.
> - Flow of Section 2, use of pseudocode, ...
> > We will improve the exposition of Section 2 in the revised manuscript following the reviewer's suggestions. With regard to parallel implementation of (12), we were considering a scenario where we can take advantage of multi-core CPUs or GPUs that each agent may possess.
> - Implementation of prox oracles
> > In the revision we will point out that prox operators are known for many common regularizers (e.g., $\ell_1$ norm and squared $\ell_2$ norm).  The prox-like operator in (12) can also have closed-form in some special cases, but the implementation of (12) is less critical as it reduces to univariate strongly convex minimization problems.
> - Tuning the hyperparameters
> > We will consider this suggestion in the revised manuscript, if the calculations show a significant gain.
> - Attributes of the communication graph and their effects
> > While we can compare the trends of the empirical results and the predictions of the theory, a more precise comparison requires experiments of much larger scale which are beyond the scope of this work. Also, the graph attributes that appear in the bounds can certainly be approximated, but in some cases (e.g., the random graph models) that would require addition of nontrivial statistical analysis, which diverges from the main theme of this work.

---

### Decision · Action_Editors · 2022-12-02

**Recommendation:** Reject

**Comment:**

This paper studies a *potentially* interesting problem, but fails to give proper justification for the studied setup. In other words, while I believe the problem *can* be well motivated, the authors did not do so to a degree that would be deemed sufficient by the reviewers and myself as well.

In this regard, one reviewer wrote: "The setting considered in the paper is motivated by applications such as federated learning and parallel computing. When motivating the applications in federated learning, the authors simply mention the cross-silo setting, without explaining how the specific issues addressed in their paper would be helpful there." Another reviewer wrote: "I concede with the other reviewers that the oracle considered in the paper is not very well motivated. And even for the given oracle, the authors are unclear about the final oracle complexity and implemented algorithm (in terms of parallelism). One could expect the authors to make some changes to improve these issues, but overall the paper will need more than one round of changes to be acceptable."

Moreover, the authors seem to be unaware of large swaths of related literature, mainly related to distributed variants of coordinate descent (there are more works on this topic than the one that was missed and pointed out by one reviewer; another example: "Fast distributed coordinate descent for minimizing non-strongly convex losses", MLSP 2014), and vertical FL (there is a very large body of recent work in this area). Their results need to be contrasted to what is known so that the readers can appreciate them in context. This is a basic requirement of the scientific method.

Besides these issues, the contribution of this paper was described by one reviewer as "extremely small". The derivation done in the paper merely amounts to dualizing the problem (a standard technique), and subsequently applying a known method (Chambolle-Pock) to the saddle-point reformulation.This can be viewed as contribution that is closer to an "exercise" difficulty level than to a "substantial research" difficulty level. This could in principle have been somewhat alleviated if the authors provided additional insights, guidelines and contributions besides this, but the reviewers concluded the paper lacks in this kind of development as well. So, the reviewers and myself find it hard to see where the original contributions are, and how they relate to the literature. Because of this, readers could be easily misled rather than enlightened.

After a short discussion, none of the reviewers proposed acceptance. I concur with these views, and therefore have no choice but to propose rejection. However, I encourage the authors to keep developing these ideas further. I believe that a very substantial revision, with more work invested into the research, could eventually lead to a nice paper.

AC

**Audience:**

Yes, but the presentation has serious issues, and the confusion that would be caused outweighs the benefits. The paper needs to address the issues raised in the reviews.

**Claims And Evidence:**

Yes, the claims are supported by evidence. However, they seem to be very simple observations and applications of known results, and hence seem not to be of a sufficient depth to warrant a publication.

---

> ### Author Response · Authors · 2022-12-19
> **Rejoinder**
>
> In view of the final decision letter, we feel compelled to clarify a few points and provide a different perspective.
>
> With regards to the applications in cross-silo federated learning (FL), we find it evident that partitioning of the data along the features is the common setup for these FL problems. In such circumstances, by being succinct, we try to avoid the pretense of sophistication through prolixity.
>
> Regarding the Vertical Federated Learning (VFL) literature, while the reviewers did not mention any missed VFL paper specifically, the "recent literature" that we have found have appeared after our first draft was posted on arXiv on October 2021. Regardless, we agreed to cite and discuss the recent VFL papers as well.
>
> With regards to the coordinate descent methods, as we mentioned in a previous response, they are certainly relevant and will be discussed in the next revision. However, our previous comment still stands: the specifically mentioned references do not consider the case of convex and Lipschitz loss functions; they assume (2nd order) smoothness.
>
> We also don't find the comments we received at least adhere to the philosophy behind the creation of the TMLR (https://www.jmlr.org/tmlr/acceptance-criteria.html).